# Playing Closer: Using Virtual Reality to Measure Approach Bias of Internet Gaming Disorder

**DOI:** 10.3390/bs13050408

**Published:** 2023-05-14

**Authors:** Wen Wei, Qi Wang, Ruyi Ding, Rui Dong, Shiguang Ni

**Affiliations:** 1International Graduate School at Shenzhen, Tsinghua University, Shenzhen 518055, China; 2Department of Psychology, Sun Yat-sen University, Guangzhou 510006, China; 3School of Business Administration, Zhejiang University of Finance and Economics, Hangzhou 310018, China

**Keywords:** internet gaming disorder, approach bias, virtual reality, measurement

## Abstract

Playing online games is gradually becoming mainstream entertainment, but some people may develop Internet gaming disorder (IGD). Like other behavioral addictive diseases, one of the main characteristics of IGD is a craving for games, which also makes people tend to approach game-related clues. Recently, a few researchers have started to use the approach–avoidance task (AAT) paradigm to study the approach bias of IGD, and they also think it is an essential characteristic of IGD. However, the traditional AAT cannot provide realistic approach–avoidance behavior to stimuli, and virtual reality has been proven to provide a highly ecological environment to measure approach bias. Therefore, this study innovatively integrates virtual reality and the AAT paradigm to measure the approach bias of IGD. We found that compared with neutral stimuli, IGD spent less time approaching game-related stimuli, which indicates that it is difficult for IGD to avoid game-related situations in the virtual environment. This study also revealed that game-related content stimuli in virtual reality alone did not increase the IGD group’s craving for games. These results proved that AAT in VR could cause the approach bias of IGD and provide high ecological validity and an effective tool for the intervention of IGD in the future.

## 1. Introduction

In recent years, online games have become a popular source of entertainment for modern people. Due to the advantage of being easy to learn, online games have also led to more people becoming addicted. The Diagnostic and Statistical Manual of Mental Disorders, Fifth Edition (DSM-5) includes Internet gaming disorder (IGD) as a disease that needs further research [1]. IGD is defined as long-term addiction to online games, resulting in significant impairment of social and psychological functioning. A review has shown that the prevalence of online game problem groups in China is about 3.5–17%, higher than the global level [2]. Given that online game addiction can lead to poor academic performance, cognitive dysfunction and impaired social interaction, it is imperative to explore the addictive characteristics of IGD and provide a more effective means of measuring these features [3].

In the DSM-5, IGD is the second formally nominated behavioral addiction disorder after pathological gambling (PG) [4]. Its clinical characteristics are similar to those of behavioral addictions, such as craving and impulsivity. Impulsivity is easily activated by addiction-related cues, which automatically drive them to approach addiction-related stimuli. This behavioral tendency is also called approach bias [5]. Some researchers have also argued that this automatic activation of motivational orientations toward addiction-related cues plays a crucial role in the association with addictive behavior [6]. The dual-process model of addiction supposes that behavioral decision making in healthy individuals is controlled by both the impulsive processing system and the reflective processing system [7]. The reflective processing system is dominated by the prefrontal cortex system, which provides rational behavioral responses considering long-term benefits. The impulsive processing system is regulated by the amygdala system, which controls impulsive behavioral responses based on immediate feedback [8]. For individuals with addictive disorders, repeated addictive behaviors inhibit or impair their automatic processing system, and the impulsive processing system becomes overactive instead. Several studies have shown that approach bias is associated with addiction development and maintenance [9]. The level of approach bias is positively correlated with the severity of the addiction. Thus, measuring the approach bias of IGD has essential theoretical and practical value.

The most common paradigm for measuring approach bias is the approach–avoidance task (AAT). Two studies demonstrated that people with IGD have approach bias toward game stimuli using a joystick and a manikin in AAT, respectively. In the joystick study, 38 participants addicted to massively multiplayer role-playing games (MMORPGs) were randomly grouped and asked to push and pull the joystick to avoid MMORPG-related stimuli [10]. The study found that people with IGD are more inclined to approach addiction-related clues. In another variant of the AAT, researchers instructed participants to manipulate a manikin on the digital screen toward or away from the center [11]. This study found that the response time to gaming stimuli in the approach condition was significantly shorter than that to neutral stimuli. However, measuring approach bias with a joystick or with a manikin in traditional AAT may not be as sensitive because individuals themselves cannot approach or avoid the stimulus [12]. To overcome these shortcomings, approach bias needs to be explored by incorporating objective measurements.

Many studies have gradually emphasized that virtual reality (VR) shows the potential to contribute to objectivity and reliability in psychiatric assessment and has steadily become a means of future psychological assessment [13]. A systematic review has shown that VR provides benefits in evaluating behavioral addictions and achieves high ecological validity [14]. VR simulates the real world by using many sensory inputs and scenarios to create a fully immersive experience [15]. This virtual environment, especially substance use, can facilitate social interactions and further stimulate the individual’s presence [16]. Additionally, the more participants feel immersed in virtual scenarios, the more this ecological approach-and-avoidance behavior activates the corresponding neuropsychological system [17]. Therefore, VR may provide a more objective assessment tool for approach bias.

One previous study fused VR and AAT (virtual approach–avoidance task—VAAT) to measure the approach bias for alcohol use disorder in VR [12]. They found that heavy drinkers had difficulty avoiding alcohol-related scenarios in a virtual environment. This research assumed that the VAAT might be a more accurate measure of social drinkers’ craving for alcohol. It allows individuals to be immersed in virtual environments and provides more realistic situations. This suggests that it is possible to measure approach bias objectively using the VAAT paradigm.

However, there are few VR studies in the field of IGD for assessment. To our knowledge, only one study constructed a virtual Internet cafe to assess the effect of VR in the IGD group [18]. In that study, participants first followed a preset pathway to enter the Internet bar. The waiter welcomed them and let them look around. The casual observation task observed their peers discussing game updates or competitions. In the game invitation task, peers in the virtual scenario invited participants to play the game together for different reasons. Participants were instructed to accept or refuse their invitation to play games together. It was found that the participants’ desire to play games was significantly higher when they entered an Internet cafe and during the game invitation task than during the everyday observation task. This study demonstrated the effectiveness of VR in eliciting game-related cravings in the IGD group. However, these two aspects may need further discussion and improvement.

The first point is that the virtual scenes used in this study included social factors such as peer pressure and interaction with the counter clerk. No distinction was made between content craving and social craving for game content itself [14]. Therefore, this study’s scenario was too complex to determine that the game stimuli in VR alone would be sufficient to elicit game craving in those with IGD. The second point is that Shin et al. [18] used self-reports rather than direct responses to addiction cues to measure hunger. As mentioned above, such self-reports can easily be exaggerated and exposed to social expectancy bias [19], so self-reported craving may not appropriately reflect the addiction degree in IGD. Using more sensitive behavioral data (e.g., approach bias) can provide a more objective assessment of addiction [20].

The present study aimed to overcome the limitations of traditional assessments by using virtual reality to objectify the measurement of IGD and an implicit task to assess automatic approach behavior towards gaming cues. Based on the empirical findings and theoretical models outlined above, we hypothesized that the IGD group would show an approach bias towards gaming cues in VR. Additionally, we aimed to study whether game content stimuli in virtual scenarios alone could increase self-reported craving for gaming cues in IGD. The present study hopes to examine further whether VR-induced approach bias will make it easier to predict the severity of Internet gaming addiction than self-reported craving.

## 2. Materials and Methods

### 2.1. Study Participants

We recruited college students from Sun Yat-Sen University by distributing questionnaires online and offline. Given that the most common type of Internet game played by those with IGD in recent years has been multiplayer Internet battle arena (MOBA) games, we only recruited participants who had recently played “Honor of Kings”, a famous mobile MOBA game in China, for at least one year [11]. It can be described as a simplified mobile version of “League of Legends” because of its high similarity to its competitive format and game content.

The inclusion criteria for IGD were (1) meeting at least five of the nine DSM-5 diagnostic criteria for IGD [21] and having an online game addiction score of 30 or more; (2) having mobile games (cell phones, iPad, etc.) as the most recent media commonly used for playing online; and (3) time spent playing “Honor of Kings” exceeding 10 h per week. Exclusion criteria included: (1) self-reported history of any underlying psychiatric disorders (e.g., depression, schizophrenia, etc.); (2) current use of any psychotropic treatment or medication; and (3) reports of any gambling experience or current or former use of illegal substances.

To control for participants with similar familiarity with game stimuli and overcome some limitations of non-gamer participants, recreational game users (RGUs) were selected as a control group in this study [22]. Compared with healthy controls, RGUs were more comparable to the IGD group regarding Internet game exposure levels. The inclusion criteria for RGUs were (1) meeting at most four of the DSM-5 diagnoses of IGD and having an online game addiction score of 24 or less; (2) reporting that the most recent media used to play online games were mobile games; (3) playing “Honor of Kings” at least five days a week and at least two hours a day. The exclusion criteria for RGUs were the same as for the IGD group.

We used G*power 3.1 to conduct an a priori sample size analysis [23]. With an alpha level of 0.05, a power of 0.95, and an assumed correlation between repeated measures of *r* = 0.50, the total sample size of at least 36 valid participants was sufficient to obtain a moderate effect size (Cohen’s d = 0.25) for the interaction.

Based on these criteria, 40 participants were selected for this study. These included 22 participants in the IGD group (12 males, 10 females) and 18 participants in the RGU group (9 males, 9 females). These participants had normal or corrected vision, and their basic information is shown in Table 1. This study was also approved by the Ethics Committee of the Department of Psychology at Sun Yat-Sen University. After receiving a detailed explanation of the study, participants provided informed consent before participation. The participants all volunteered to participate in the experiment and received financial compensation.

### 2.2. Measures

#### 2.2.1. Internet Gaming Severity

The IGD severity scale was developed by Petry et al. [24] to measure the severity of Internet gaming addiction (e.g., Do you feel you should play less, but have failed to reduce the amount of time you spend playing games?). Participants rated every item on a 5-point Likert scale that ranged from 1 (very inconsistent) to 5 (very consistent). Its internal consistency coefficient was 0.84 in this study.

#### 2.2.2. Weekly Gaming Time

The weekly time spent on the game was calculated by multiplying the results of the following two questions [11]: (1) the average number of days spent playing “Honor of Kings” per week based on 8 Likert points (1 = “Never” to 8 = “Almost every day”) and (2) the average game time spent playing “Honor of Kings” each time, using 7 Likert points (1 = “<30 min” to 7 = “8 h or more”).

#### 2.2.3. Subjective Gaming Craving

Subjective game craving was measured using a 10-point Likert rating scale (0 = “do not want to play at all” to 10 = “want to play very much”) [25]. This has recently been frequently used to assess IGD craving for games [18].

#### 2.2.4. Depression Symptoms

The Beck Depression Inventory measured depression symptoms [26]. The scale consists of 21 items, each rated from 0 to 3 (e.g., 0 = I do not feel sad; 1 = I feel sad a lot of the time; 2 = I feel sad all the time; 3 = I am too sad or too upset to bear). The higher the total score, the more severe the depression symptoms. The Cronbach’s α coefficient in this study was 0.79.

#### 2.2.5. Presence Questionnaire

The sense of presence scale uses the Chinese version of the Independent Television Commission-Sense of Presence Inventory (ITC-SOPI), first developed by Lessiter et al. [27]. It has 44 entries and includes four factors that affect the sense of presence: engagement (e.g., I felt sad that my experience was over), ecological validity (e.g., The displayed environment seemed natural), spatial presence (e.g., I felt I could interact with the displayed environment), and adverse effects (e.g., I felt disoriented). This Chinese version of the ITC-SOPI has been shown to have good psychometric properties and reliably and effectively measure presence using VR [28].

### 2.3. Procedure

In the virtual approach–avoidance task, participants first sat on a chair and wore a VR device to familiarize themselves with the virtual environment. The virtual environment was a private bedroom familiar to university students. They were told they needed to treat the current scene as their bedroom. After familiarizing themselves with the scene, the participants held the joystick and used it to pick up the electronic device on the virtual table. They were asked to adjust the distance between their eyes and the device until it was appropriate. After the participant was prepared, they pulled the trigger and started the experiment.

In the training phase, the participant needed to respond to the signal according to the following instructions. When each video clip is played, if the border of the electronic device in front of the participant turns red, the participant should pull the joystick as soon as possible; when the edge turns blue, the participant should push the joystick as quickly as possible. After each response, the participant will receive feedback. When the feedback is finished, the participant needs to return to their original position. After eight correct responses, the participant will begin the formal experiment.

The formal VAAT experiment included 2 blocks with 48 trials. Each block used a hybrid design of 2 (instruction: push, pull) × 2 (context: game, neutral), with 6 video stimuli for each scenario. In each trial, game- and non-game-related video clips were first presented on a virtual electronic screen, with each video presentation lasting approximately 8s. Afterward, the participants were required to push or pull according to the electronic device border color. In the push condition, participants manipulated the virtual arm by pushing the joystick to avoid stimuli. In the pull condition, participants used the virtual arm by pulling back the joystick to move closer to the inspiration in the virtual environment. The participants received feedback after each response and moved on to the subsequent trial 3000 ms after feedback was given. Video clips and border colors were randomly selected to avoid participants’ anticipation of the stimulus’ appearance.

The game-related video stimuli were selected from the appearance animation of heroes in “Honor of Kings”, and the related control stimuli were from the introduction of heroes’ appearances in infamous cartoons. The control stimuli were similar to the game clips in shading, sound effects, narration, and weapons, and the genders remained consistent. Therefore, all participants had the experience of watching the relevant animation of the control stimuli, and the cartoon and the hero did not appear in any online game.

The dependent variable recorded in the experiment was the reaction time for participants to make a correct response after the border of the virtual electronic device changed color. This study used median reaction times rather than mean scores to minimize the effect of outliers, and it was also not necessary to define a cutoff point for extreme values [11]. The experiment recorded the distance between the handle and the helmet immediately after the video clip was played. It did not determine whether the push or pull was completed until the participant moved away from or tended toward the participant’s eyes beyond the initially recorded distance and gave immediate feedback accordingly. In this study, we recruited two participants, one male and one female, for the pre-experiment to determine the appropriate convergence and divergence intervals. After adjusting several rounds of parameters until the participants felt that the feedback of pushing or pulling was sensitive, a Euclidean distance of plus or minus 5 cm was selected as the interval for judging pushing or pulling. As shown in Figure 1, when the participant saw the red border, pulling the joystick from the original position toward the eye would immediately be judged as correct, and pushing the joystick beyond the interval would be considered wrong.

HTC Vive was used as the VR system in this study. This VR system is run with the following PC specifications: Lenovo Y720, CPU Intel^®^ Core™ i7-7700, Windows 10, and GPU NVIDIA^®^ GeForce^®^ GTX 1070. After signing the informed consent form, each subject underwent basic demographics and measured their subjective craving to play “Honor of Kings”, and then completed the Online Game Addiction Scale. Afterward, the participants conducted the entire VAAT experiment, which lasted approximately 20 min. After completion, each participant was instructed to take the subjective gaming craving measure again and complete the Presence Scale and the Depression Scale.

### 2.4. Data Analysis

Independent sample t-tests were first used to compare differences between IGD and RGU demographics, online game addiction, and VR-related variables. Approach bias between the two groups was analyzed using a three-factor repeated-measures analysis of variance (ANOVA). If the three-way interaction was significant, a two-factor repeated-measures ANOVA was used to explore the interaction between stimulus type and response instructions in each group.

A 2 × 2 mixed-measures ANOVA (IGD and RGU as between-group variables; pretest and posttest as within-group variables) was used to analyze the interaction and main effects of the level of craving for the game. Independent sample *t*-tests were conducted to compare the differences in subjective craving between the IGD and RGU groups before and after the experiment.

Approach bias scores were obtained by subtracting the individual’s approach bias scores: approach bias toward gaming cues = time spent avoiding online game cues − time spent approaching online game cues; approach bias toward control cues = time spent avoiding control cues − time spent approaching control cues. A positive score indicates a convergence trend, and a negative score reflects an avoidance trend. A higher score indicates a more pronounced approach bias toward a particular stimulus. The change in subjective game craving is obtained by subtracting the game craving measured after the VAAT from the one measured before. A positive score indicates an increase in game craving and vice versa. The higher the score, the more the desire for games increases.

In this study, Pearson linear regression was used to investigate whether a correlation exists between online game addiction and approach bias scores and changes in subjective game craving. Finally, multiple linear regression was used to examine which factors were significantly correlated with online game addiction scores. This was carried out using game approach bias scores, weekly gaming time, and depressive symptoms as independent variables.

## 3. Results

### 3.1. Demographic and Clinical Characteristics of IGD versus RGU

The statistical variables, IGD scores, depressive symptoms, and VR scale scores of each group were first analyzed using descriptive statistics, and the between-group differences for each variable were explored. The results are shown in Table 1. The results showed significant differences between IGD and RGU in the degree of online game addiction (*t*(38) = 12.10, *p* < 0.001) and the number of hours played per week (*t*(38) = 2.56, *p* < 0.05).

As shown in Table 2, both weekly gaming time and IGD scores were significantly positively correlated with the degree of depressive symptoms (*r* = 0.42, *p* < 0.05; *r* = 0.37, *p* < 0.05). Spatial perception and engagement were significantly positively correlated (*r* = 0.60, *p* < 0.01). Age and negative effects were significantly negatively correlated (*r* = −0.37, *p* < 0.05).

### 3.2. Three-Way Repeated-Measures ANCOVA

Three-factor repeated-measures variance results revealed a significant three-factor interaction between group, stimulus type, and response instruction (*F*(1, 38) = 4.49, *p* < 0.05, *η*^2^ = 0.11) and significant main effects for stimulus type and response instruction (*F*(1, 38) = 137.72, *p* < 0.001, *η*^2^ = 0.78 and *F*(1, 38) = 33.24, *p* < 0.05, *η*^2^ = 0.47, respectively). However, the interactions between group and response instruction and group and stimulus type were not significant. In addition, when depression level was included as a covariate, a three-way repeated-measures analysis of covariance (ANCOVA) found no significant interactions between group, stimulus type, and response instruction (*F*(1, 37) = 2.10, *p* = 0.16, *η*^2^ = 0.05).

A two-factor repeated-measures ANOVA was conducted to explore the interaction between stimulus type and response instructions in each group, and the results are shown in Figure 2. For the IGD group, the interaction between stimulus type and response instructions was significant (*F*(1, 21) = 4.78, *p* < 0.05, *η*^2^ = 0.19), and there were also significant main effects for stimulus type and response instructions (*F*(1, 21) = 42.38, *p* < 0.001, *η*^2^ = 0.67 and *F*(1, 21) = 27.00, *p* < 0.001, *η*^2^ = 0.56, respectively). Simple-effects analysis showed that the IGD group had significantly faster reactions to game stimuli than control stimuli when following pulling instructions (*F*(1, 21) = 15.77, *p* < 0.001, *η*^2^ = 0.44). In contrast, the interaction of both stimulus type and response instructions was insignificant in the RGU group.

### 3.3. Differences between Game Craving Pre- and Posttest

The level of craving for the game was analyzed using a 2 (time: pretest, posttest) × 2 (group: IGD, RGU) ANOVA, and the results showed that the time and group interactions and the time main effect were not significant, but the group main effect was significant (*F*(1, 38) = 6.82, *p* < 0.05, *η*^2^ = 0.15). Independent sample t-tests were applied to game craving in each group before and after the experiment. The results showed that the IGD group was significantly higher than the RGU group both before and after the experiment (pretest: *t*(38) = 2.55, *p* < 0.05; posttest *t*(38) = 2.13, *p* < 0.05), as shown in Figure 3.

### 3.4. Exploratory Analysis of the Severity of Internet Game Addiction

For exploratory analysis, game approach bias, weekly game playing hours, and depression symptoms were put together in a multivariate linear model to investigate which factor could be significantly associated with online game addiction scores in the IGD group. The results showed that these three variables explained 52.6% of the variance (adjusted *R*^2^), and only game approach bias was significantly and positively correlated with online game addiction scores (Table 3, Beta = 0.30, *p* < 0.05).

## 4. Discussion

The primary purpose of this study was to determine if the VR environment could elicit IGD to approach game cues. Three results were obtained from this study. First, the IGD group spent less time approaching game-related stimuli in a virtual environment. Second, game-related content stimuli in virtual scenes alone did not increase the IGD group’s craving for game stimuli. Third, the increase in the approach bias of gaming stimuli predicted the degree of online game addiction. These results suggest that using the AAT paradigm in a virtual environment can be used more sensitively to measure approach bias and provide a research basis for subsequent IGD treatment interventions.

The results showed that the IGD group spent less time approaching the game-related stimulus than the control stimulus. This is consistent with previous findings indicating that the IGD group tended to approach online game cues [10]. The results also suggested that approach bias toward games is an essential feature of IGD [11], which is also prevalent in IGD and other addiction disorders. Additionally, both the IGD and RGU groups have faster response times than control stimuli. This suggests that both have some degree of preference for game stimuli and that this preference speeds up their responses to some extent. This conditioned processing of stimuli provides strong evidence supporting that IGD is a behavioral addiction, suggesting that rapid, unconscious, and habitual behaviors may be associated with the formation of IGD [29]. At the same time, this approach bias can be considered a result of an imbalance between the automatic processing system and the impulsive processing system, and the former may be dominant in IGD [12].

In the present study, although the subjective craving for games was significantly higher in the IGD group than in the RGU group, it failed to successfully elicit emotional levels of desire for games with game content stimuli in both IGD and RGU. Shin et al. [18] found that a game invitation task produced greater craving than a session observation task. Gao et al. [30] illustrated through a questionnaire that cyber cafes and game halls are more likely to induce IGD as the primary environment for offline communication among online gaming players than in other environments. These studies showed that social factors are more likely to be essential components in the development of craving in the IGD group. In contrast, content factors may have less influence on game craving. More social interaction in a virtual environment may be successful at increasing IGD craving levels. Therefore, this study also calls on future researchers to distinguish between social and content craving when exploring subjective desire and examine the effects of different factors on eliciting passion separately.

For exploratory analysis, we found that the increase in the approach bias of gaming stimuli predicted the degree of online game addiction. This suggests that approach bias may reflect online game addiction more readily than weekly gaming time and depressive symptoms. According to the stimulus sensitization theory, repeated exposure to games increases automatic processing. This conditioned stimulus, as an incentive, captures one’s attention and increases behavioral motivation for substance cues in the addicted group [4]. These suggest that individuals with IGD are easily influenced by game-related cues in terms of impulsivity. Our study indirectly showed that approach bias is an important feature of IGD, which may be related to the occurrence, development, and maintenance of IGD [9].

For the ecological nature of VR, we did not find differences between the IGD and RGU groups on each subscale of the Presence Scale. However, our study indicates that the constructed VR scenarios did not cause too much discomfort to the participants while having high ecological validity. This further suggests that VR technology can be an appropriate measurement of IGD. For example, some researchers have demonstrated that an approach–avoidance training program (AATP) using VR can effectively treat alcohol use disorder and is superior to traditional PC-based AATP at three-month follow-up [31]. They found that VR-based AATP could be implemented easily and cheaply as an add-on treatment or continued care to enhance the effectiveness of current evidence-based treatment. He et al. [11] explored the modification effect of AATP training on a 2D digital monitor for the IGD group, proving that AATP can mitigate IGD severity by decreasing IGDs’ cognitive bias toward Internet gaming cues. Future researchers may explore whether AATP in VR is equally effective for the IGD population and whether it has advantages over conventional AATP. In addition, group behavioral interventions combining VR with positive meditation have also been shown to reduce decision-making impulsivity in the IGD group [32]. Future research could investigate whether their combined therapy can reduce approach bias and online game severity in the IGD group.

In line with previous studies, this paper also confirmed an association between online game addiction and depression among male and female players [33]. To summarize the factors discussed above, social factors such as adverse peer influence, lack of social support, and poor psychological status may be risk factors for the development of RGU to IGD [30]. Therefore, we call for the early screening of the RGU population and the determination of psychological assistance to reduce their risk of developing IGD. It is also important not to neglect examining the depressive status of those diagnosed with IGD for the presence of comorbidities and providing appropriate treatment options.

There are several shortcomings in this study. First, the number of participants was relatively small. Only 40 participants were selected for this study. Although studies published in this field also have similar sample sizes [11,12], a larger sample size could better represent the IGD population and make the results more reliable. This study was also conducted only with one type of game, which may affect the external validity of the study for other game types. Second, we diagnosed IGD using scales and did not use structured interviews or professional clinicians to determine the IGD group. Third, the participants recruited in this study often played mobile games, and no further distinction was made between online game addiction and mobile game addiction. While some researchers found that the mobile game addiction group is larger than the online game addiction group [34], future researchers can further explore the difference and the connection between the two groups. Furthermore, we see that the IGD and RGU groups also differ significantly in their playing time and there is a highly significant difference in their IGD scores, meaning that perhaps there may have been other variables that influenced the difference between the two groups. Future studies should explore larger samples and consider controlling for the level of proficiency and familiarity with the game to explain why some individuals tend to crave gaming more than others.

## 5. Conclusions

This study demonstrated for the first time that IGD showed approach bias toward Internet gaming cues in VR, further supporting the basis for IGD as a behavioral addiction. This study also found that those game content stimuli alone did not cause the increase in the IGD group’s craving for game stimuli, calling for future researchers to distinguish between social craving and content craving when exploring craving levels. In addition, the VR scenario adopted in this study was also shown to have better spatial perception and engagement. This suggests that approach bias might be measured more ecologically using the improved VAAT paradigm. Future research can further characterize approach bias as a behavioral feature of IGD while developing VR interventions for IGD.

## Figures and Tables

**Figure 1 behavsci-13-00408-f001:**
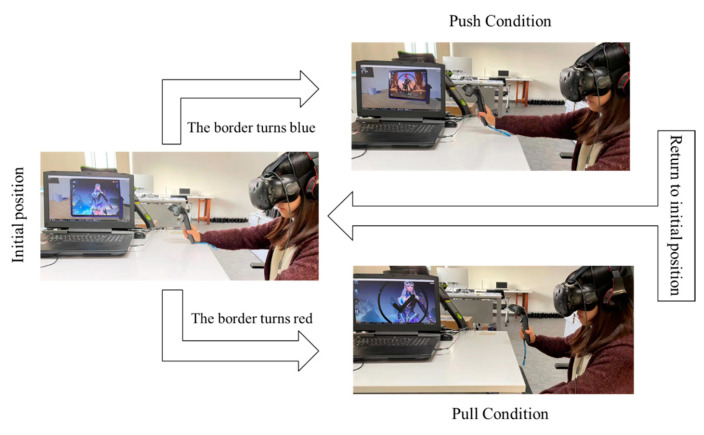
Schematic diagram of the participant wearing a VR device to complete the VAAT.

**Figure 2 behavsci-13-00408-f002:**
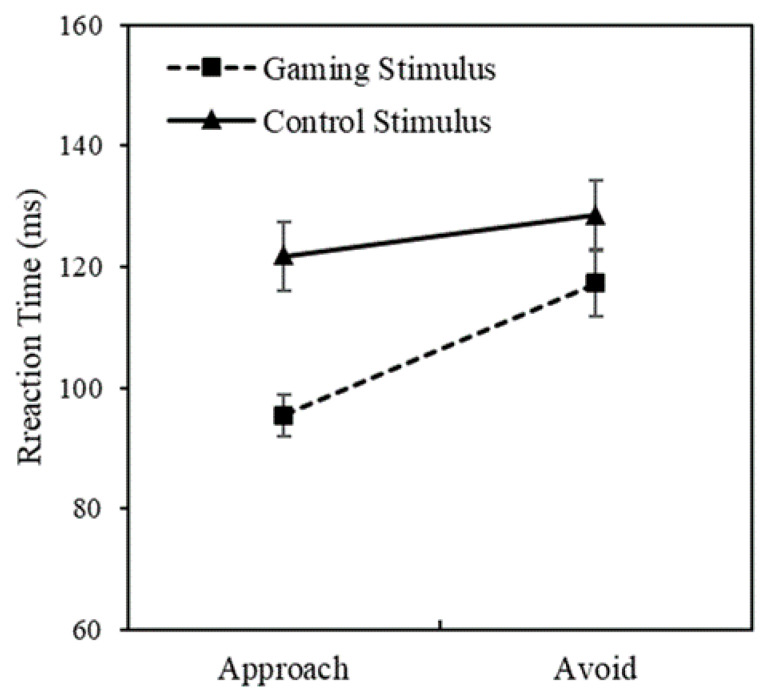
Differences in reaction times of the IGD group toward different stimuli and instructions.

**Figure 3 behavsci-13-00408-f003:**
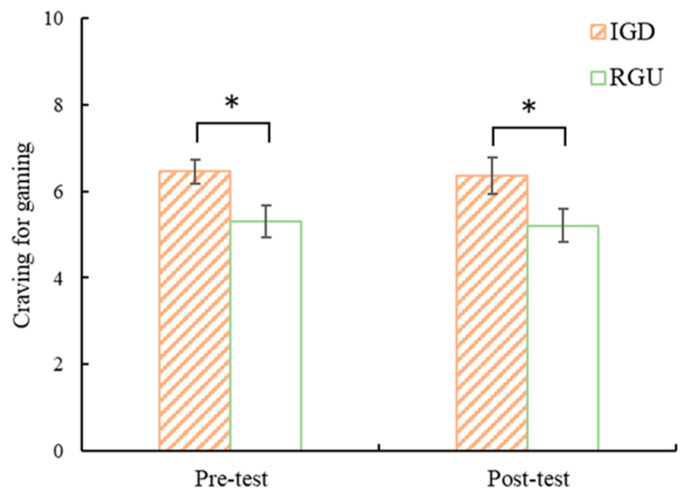
Pre- and post-test comparison of the IGD and RGU groups’ craving levels for games. Note. * *p* < 0.05.

**Table 1 behavsci-13-00408-t001:** Demographic characteristics of IGD and RGU groups and comparison of each variable.

Variable	IGD (*n* = 22) (Mean ± SD)	RGU (*n* = 18) (Mean ± SD)	*t*	*p*-Value
Gender	-	-	−0.28	0.78
Age (years)	20.41 (1.89)	19.88 (1.90)	0.82	0.42
Years of education	16.27 (2.64)	15.70 (1.57)	0.42	0.68
Weekly gaming time	15.36 (3.59)	12.89 (1.90)	2.56	0.02
IGD scores	33.77 (2.33)	20.61 (3.50)	12.10	<0.001
Depressive symptoms	11.41 (6.95)	7.67 (4.20)	2.00	0.05
Spatial presence	3.36 (0.39)	3.33 (0.57)	0.15	0.88
Engagement	3.37 (0.43)	3.28 (0.54)	0.60	0.55
Ecological validity	3.21 (0.67)	3.29 (0.58)	−0.40	0.70
Negative effects	2.75 (0.65)	2.66 (0.65)	0.36	0.72

**Table 2 behavsci-13-00408-t002:** Correlation analysis among variables.

Variable	1	2	3	4	5	6	7	8	9
1 Age	1								
2 Years of education	0.78 **	1							
3 Weekly gaming time	0.25	−0.21	1						
4 IGD scores	0.04	0.02	0.36 *	1					
5 Depressive symptoms	−0.12	−0.07	0.42 *	0.37 *	1				
6 Spatial presence	−0.07	−0.10	−0.14	0.04	0.11	1			
7 Engagement	0.16	0.01	0.02	−0.02	0.04	0.60 **	1		
8 Ecological validity	0.22	0.17	−0.01	−0.06	−0.13	0.12	0.24	1	
9 Negative effects	−0.37 *	−0.21	0.09	0.20	0.29	0.04	−0.16	−0.08	1

* *p* < 0.05, ** *p* < 0.01.

**Table 3 behavsci-13-00408-t003:** Multiple linear models of standardized coefficients of the IGD scores in the IGD group.

Variable	Beta	*p*-Value
Game approach bias	0.303	0.045
Weekly gaming time	0.235	0.141
Depressive symptoms	0.206	0.203

## Data Availability

The data presented in this study are available on request from the corresponding author.

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
