# Peer review of "Playing Closer: Using Virtual Reality to Measure Approach Bias of Internet Gaming Disorder"

_behavsci, 2023, doi:10.3390/bs13050408_

Round 1

Reviewer 1 Report

The study successfully demonstrates that using the AAT paradigm in a virtual environment can be used more sensitively to measure approach bias and provide a research basis for subsequent IGD treatment interventions. The three primary results from this study are also promising. Firstly, the IGD group spent less time converging on game-related stimuli in a virtual environment. Secondly, game-related content stimuli in virtual scenes alone did not increase the IGD group's craving for game stimuli. Thirdly, the increase in the approach bias of gaming stimuli predicted the degree of online game addiction. Strong evidence is provided supporting IGD as a behavioral addiction, suggesting that rapid, unconscious, and habitual behaviors may be associated with the formation of IGD. However, it is important to distinguish between social and content craving when exploring subjective desire and examine the effects of different factors on eliciting passion separately. I suggest that future researchers may explore whether AATP in VR is equally effective for the IGD group. As well, I’d suggest one more review for clear English grammar, but these would be minor revisions for clarity. 

Author Response

Point 1: I suggest that future researchers may explore whether AATP in VR is equally effective for the IGD group.

Response 1: We would like to thank you for your careful reading and helpful comments. We agreed that knowledge is lacking as to whether the effectiveness of AATP can be enhanced further in VR when performed for the IGD group. Our study first demonstrated that AAT in VR could cause the approach bias of IGD and the stimuli-relevant AAT seemed particularly suitable for implementation in a VR format. Mellentin et al. (2020) supposed that the VR format of AATP had the potential to make the paradigm even more so by navigating in a relatively interactive environment and approaching or avoiding related stimuli in a wider sense. We have added more discussion on this point in the Discussion section and also hope that future researchers may examine the effect of VR-based AATP in the longer term.

Point 2: As well, I’d suggest one more review for clear English grammar, but these would be minor revisions for clarity. 

Response 2: Thank you so much for your careful check. We have re-examined the grammar of this article and corrected the errors we found in our revised manuscript.

Reviewer 2 Report

The paper titled "Playing Closer: Using Virtual Reality to Measure Approach Bias of Internet Gaming Disorder" presents an innovative approach to measure the approach bias of Internet Gaming Disorder (IGD) using the virtual reality (VR) technology. The authors argue that traditional approach-avoidance task (AAT) paradigms cannot provide a realistic approach-avoidance behavior to stimuli, which limits their ecological validity. Therefore, they propose integrating VR and the AAT paradigm to measure the approach bias of IGD.

The study finds that IGD spends less time approaching game-related stimuli compared to neutral stimuli, which suggests that IGD finds it difficult to avoid game-related situations in the virtual environment. Additionally, the study reveals that only content cannot elicit craving for games in the IGD group. The authors conclude that the AAT in VR could cause the approach bias of IGD and provide a highly ecological and effective tool for the intervention of IGD in the future.

Overall, the paper presents an interesting and valuable contribution to the literature on IGD. The integration of VR technology in measuring approach bias is a novel approach that adds to the existing literature on IGD. The study's results provide valuable insights into the cognitive mechanisms underlying IGD and could inform the development of more effective interventions.

However, the paper's narrow research design limits the generalizability of its findings. The sample size is relatively small, and the study only focuses on one type of game. Therefore, it is crucial to replicate the study with a larger and more diverse sample to assess the generalizability of the findings. Nevertheless, the paper's contribution is significant, and it could inspire future research to explore the potential of VR in measuring and intervening in addictive behaviors.

Author Response

Point 1:  However, the paper's narrow research design limits the generalizability of its findings. The sample size is relatively small, and the study only focuses on one type of game. Therefore, it is crucial to replicate the study with a larger and more diverse sample to assess the generalizability of the findings.

Response 1: We would like to thank you for your careful reading. It is really true as Reviewer suggested that the number of participants was relatively small and that participants who play other types of games should be considered. We recruited 222 participants who completed the screening questionnaire, but only 18% of them met our inclusion criteria. Given that the most common type of Internet games played by those with IGD in recent years has been MOBA games, we recruited only participants who played this type of game by referring to previous studies with Chinese samples (He et al., 2021). But, we agreed that a larger and more diverse sample size could better represent the IGD population and make the results more reliable. So, we have added more discussion on the potential limitations of the current study in the Discussion section.

Reviewer 3 Report

Overall, I think the topic and quality of this manuscript are fine. However, I do have a question about a few things.

- First, I wonder if the differences in the cognitive function of the subjects and the processing speed in the behavioral task had any influence.

- Second, isn't it possible that the game-related stimuli could simply work as more sensory-salient stimuli rather than addiction-related cues?

- Lastly, even RGUs may have a different level of proficiency and familiarity with the game than IGD.

Author Response

Point 1: First, I wonder if the differences in the cognitive function of the subjects and the processing speed in the behavioral task had any influence.

Response 1: We gratefully appreciate your careful reading and valuable comments. We admit that cognitive functioning may vary between groups and lead to differences in performance on behavioral tasks. Dong et al. (2020) found that the RGU groups showed greater executive control and greater activations of brain regions implicated in motivational processes as compared with IGD subjects. In our study, we used the independent samples t-test to analyze differences in mean reaction time between the IGD and RGU groups. We found that there was no significant effect for different groups, t(38) = 0.41, p = .686. This result revealed that the processing speed did not differ significantly between the different groups in our experimental task. This may be because the cognitive experiment in our study was relatively simple, only needed to pull or push the joystick according to the instructions.

Point 2: Second, isn't it possible that the game-related stimuli could simply work as more sensory-salient stimuli rather than addiction-related cues?

Response 2: Thank you for your rigorous consideration. For the IGD group, there was no significant difference in the subjects' craving for the game before and after the experiment. Addiction-related cues can not only increase the physical arousal of individuals with addictive disorders, but may also result in game craving. In our study, we found that game-related stimuli induced approach bias in the IGD group, while not causing a significant change in game craving. The approach bias to addiction-related cues has been confirmed in other types of addiction (e.g., nicotine use disorder, alcohol use disorder, and gambling disorder). This suggests that game-related stimuli may induce other addiction-related behaviors in addition to inducing sensory arousal and also act to some extent as addiction-related stimuli. Considering the Reviewer’s concern, we have added more discussion on this point in the Discussion section.

Point 3: Lastly, even RGUs may have a different level of proficiency and familiarity with the game than IGD.

Response 3: We totally understand the reviewer's concern. Most previous studies have focused on the differences between IGD participants and healthy controls (HC). However, the HC are non- or low-frequent game players, who have limited experience with online gaming. This was the main reason why RGU was chosen as the comparison group for our study. In our study, the results showed significant differences between IGD and RGU in the number of hours played per week (t(38) = 2.56, p = 0.02). This suggested that the proficiency and familiarity with the game in the RUG group may differ from the IGD group. In future studies, the requirements for RGU's game familiarity should be closer to those of the IGD group. Considering the Reviewer’s concern, we have revised the discussion on this point in the Discussion section.

Round 2

Reviewer 3 Report

The authors did an excellent job responding to my inquiries. I agree to the publication of this manuscript.